# Text-Driven Image Manipulation via Semantic-Aware Knowledge Transfer

## Abstract

Semantic-level facial attribute transfer is a special task to edit facial attribute, when reference images are viewed as conditions to control the image editing. In order to achieve better performance, semantic-level facial attribute transfer needs to fulfil two requirements: (1) specific attributes extracted from reference face should be precisely transferred to target face; (2) irrelevant information should be completely retained after transferring. Some existing methods locate and modify local support regions of facial images, which are not effective when editing global attributes; the other methods disentangle the latent code as different attribute-relevant parts, which may transfer redundant knowledge to target faces. In this paper, we first propose a novel text-driven directional latent mapping network with semantic direction consistency (SDC) constrain to explore the latent semantic space for effective attribute editing, leveraging the semantic-aware knowledge of Contrastive Language-Image Pre-training (CLIP) model as guidance. This latent space manipulation strategy is designed to disentangle the facial attribute, removing the redundant knowledge in the transfer process. And on this basis, a novel attribute transfer method, named semantic directional decomposition network (SDD-Net), is proposed to achieve semantic-level facial attribute transfer by latent semantic direction decomposition, improving the interpretability and editability of our method. Extensive experiments on CelebA-HQ dataset show that our method achieves impressive performance over the state-of-the-art methods.

## 1 Introduction

Generative Adversarial Networks (GANs) (Goodfellow et al., 2014) have revolutionized a variety of fields due to its powerful ability to generate realistic and meaningful outputs. Recent works (Jahanian et al., 2019; He et al., 2019; Goetschalckx et al., 2019) have shown that deep generative models can capture real-world data distribution, and encode them into a semantically-rich latent space. Inspired by this, a lot of tasks draw their attention on the latent space manipulation, including image enhancement (Ledig et al., 2017; Yang et al., 2021b), editing (Shen et al., 2020; Härkönen et al., 2020; Patashnik et al., 2021), and discriminative tasks (Nitzan et al., 2021; Xu et al., 2021).

With the tremendous success of deep generative models, facial attribute editing (Yeh et al., 2017; Liu et al., 2019; He et al., 2020; Dorta et al., 2020), aiming to edit the specific attributes of the target facial image, has become topical. As a special case of facial attribute editing, facial attribute transfer (Xiao et al., 2018; Lin et al., 2018; Yin et al., 2019; Choi et al., 2018; 2020) uses knowledge from reference image as a condition to edit the corresponding attribute from the target image. In order to ensure that the manipulated facial image meets the requirements and interests, facial attribute transfer task tackles two challenges simultaneously: (1) **editing relevance:** the relevant attribute should be edited precisely according to the given condition; and (2) **keeping irrelevance:** the irrelevant part (e.g., identify information, background, or other attributes) should not be modified during attribute transfer. Due to strong entanglement of the attributes, meeting both requirements is an intractable task. For example, without fully disentanglement, transferring the "smile" attribute to the target facial image may cause that, another irrelevant but coupled attribute, e.g. "cheek color" attribute, would be changed during editing.

In view of the above issues, recently a variety of methods explore the attribute disentanglement in two ways. Some methods (He et al., 2020; Kwak et al., 2020) resort to the way of spatial attention

detection, which disentangle the attribute by searching specific support region spatially and only manipulate the image in such a confined area. Obviously, these methods totally ignore the facial details beyond the support region when the edited attribute is a global attribute, such as "smile" or "age". Meanwhile, other methods (Shen et al., 2020; Yang et al., 2021a; Patashnik et al., 2021) pay attention to the latent space factorization through pre-trained GAN. These methods employ the high-level semantic information as guidance to manipulate image in latent spaces, which are more suitable to handle both global and local attribute editing. However, owing to over-coupled semantic features, these methods are hard to manipulate specific attribute without powerful supervision.

To overcome the problems mentioned above, we explore the latent semantic space for disentangled attribute editing, and apply the discovered manipulation method to facial attribute transfer task. As for attribute editing task, in order to disentangle and edit the attribute specified by the text prompt, we design the directional latent mapping network, which leverages semantic direction consistency (SDC) loss to constrain the manipulation in the CLIP-space (Radford et al., 2021). The key idea of the SDC loss is employing the change direction of semantic feature to estimate the latent manipulation. Furthermore, in order to apply this effective editing method to facial attribute transfer task, we propose a novel semantic-level facial attribute transfer method driven by text prompt, named as semantic directional decomposition network (SDD-Net). The SDD-Net extracts and transfers the specific attribute without redundant information through attribute-manipulated semantic directional decomposition.

Our contributions are summarized as follows:

- We propose a novel method namely directional latent mapping network for facial attribute editing, which utilizes the semantic direction consistency regularization to ensure attribute disentanglement.

- To further take advantage of semantic direction constrain, we propose a text-driven semantic directional decomposition network (SDD-Net) for semantic-level attribute transfer, by transferring the knowledge from the reference image to the target image.

- Extensive experiments on CelebA-HQ (Karras et al., 2017) dataset show that our method achieves significant improvements over the state-of-art approaches.

## 2 RELATED WORK

**Latent Space Manipulation.** Recent studies (Bau et al., 2020; Goetschalckx et al., 2019; Shen et al., 2020) have shown that numerous GAN models can encode rich crucial information in the intermediate latent space, such as $\mathcal{W}$, $\mathcal{W}+$ (Abdal et al., 2019), or *StyleSpace* $\mathcal{S}$ (Wu et al., 2021). By learning to modify the intermediate latent code, generative models can transfer attributes from one face to another face (Xiao et al., 2018; Choi et al., 2020). To find a latent code that allows for meaningful manipulation, some methods try to learn an effective encoder network, which inverts a real image into latent space: encoder4editing (e4e) (Tov et al., 2021) method presents an encoder that is specifically designed for balancing distortion-editability tradeoff and a distortion-perception tradeoff within the StyleGAN latent space; Stylespace (Wu et al., 2021) proposes a space of channel-wise style parameters that disentangle attributes by controlling attribute-related style channels; The pixel2style2pixel (pSp) (Richardson et al., 2021) method utilizes a novel encoder architecture that inverts a real image into $\mathcal{W}+$ space without optimization. Other methods mainly focus on finding such latent code modification approach as used to traverse latent space, result in the desired manipulation: AttGAN (He et al., 2019) applies attribute classification constraint to model the relation between the attributes and the latent representation; InterfaceGAN (Shen et al., 2020) decouples some entangled semantic features with subspace projection; GANspace (Härkönen et al., 2020) leverages principal component analysis to identify mainly direction in latent space; L2M-GAN (Yang et al., 2021a) imposes an orthogonality constraint to ensure disentanglement in latent space; StyleCLIP (Patashnik et al., 2021) leverages CLIP models to guide the attributes manipulation by latent semantic matching.

Different from the methods mentioned above, our method aligns the guidance direction, which is derived directly from text prompt, with the change direction of semantic features extracted by CLIP to guide the manipulation in the latent space. By constraining the consistency of these directions, our method can achieve disentangled attribute editing.

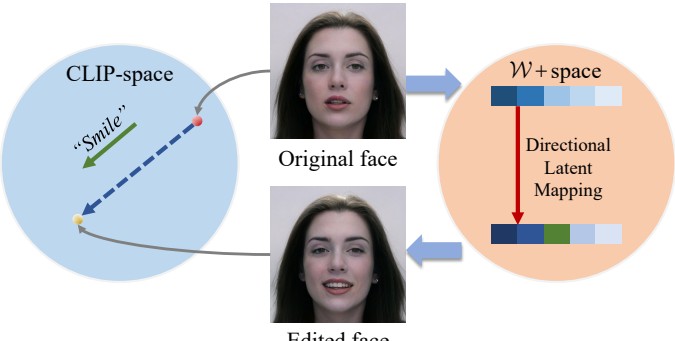

Figure 1: Overview of our idea for directional latent mapping network. By enforcing the change direction of semantic feature to align with desired direction in CLIP-space, our method can achieve impressive disentangled image manipulation.

**Semantic-level Attribute Transfer.** Semantic-level attribute transfer is a more challenging facial attribute editing task. Recent studies have attempted more detailed approaches for facial attribute transfer: StarGAN (Choi et al., 2018) applies cycle consistency to preserve identity, and uses classification loss to transfer between different domains. In addition, StarGANv2 (Choi et al., 2020) also could synthesize reference-guide images leveraging multiple domain translation. Some works are specialized for specific attributes: ExprGAN (Ding et al., 2018) proposes a model to learn the disentangled identity and expression representations explicitly for facial expression transfer. Besides, ERGAN (Hu et al., 2020) proposes a dual learning scheme to simultaneously learn two inverse manipulations for attribute transfer.

Different from these works, our proposed method focuses not only on one attribute, but also on various global or local attributes. Given an explicit text prompt, our method can automatically extract the specific attribute from the reference, and transfer the knowledge to the target image.

## 3 METHODOLOGY

In the following section, we first provide the preliminaries and problem formulation. Then, we introduce our text-driven directional latent mapping network for disentangled facial attribute editing. Finally, we describe the text-driven semantic directional decomposition network (SDD-Net) for semantic-aware facial attribute transfer.

### 3.1 PRELIMINARIES

**StyleGAN.** The StyleGAN (Karras et al., 2019; 2020) generator consists of two main components: mapping network and synthesis network. The former translates latent code $z$ to latent code $w$, which is in semantic-rich latent space $\mathcal{W}$, and the latter utilizes latent code $w$ to synthesize final images through different layers. Due to the rich information inside, the latent code $w$ can control the multi-granularity semantic features of the synthetic image, which is utilized for effective facial attribute transfer in semantic-level. In this paper, we leverage the pre-trained synthesis network as our image generator.

**StyleCLIP.** The StyleCLIP (Patashnik et al., 2021) method first combines StyleGAN and CLIP as a strong tool for text-driven image editing. The key idea of StyleCLIP is to leverage the CLIP as latent manipulation guidance. By mapping multi-modal inputs to the CLIP-space, StyleCLIP could ensure that the synthetic image matches with the text prompt in semantic-level. The objective is given by:

$$\mathcal{L}_{styleclip} = D_{\mathrm{CLIP}}(G(w), text), \tag{1}$$

where $D_{\mathrm{CLIP}}$ is the cosine distance metric in CLIP-space, and $G$ is the pre-trained StyleGAN generator.

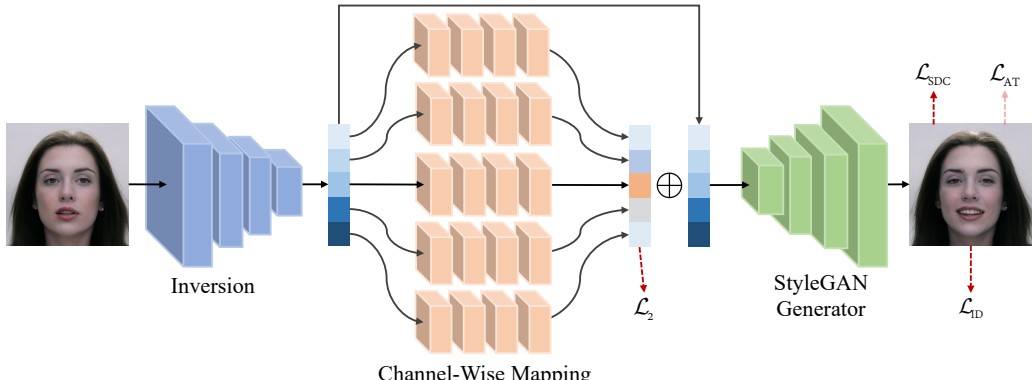

Figure 2: The architecture of our directional latent mapping network. Given the text prompt, we manipulate the latent code inverted by the facial images. Then we use $g$ group channel-wise mapping networks to manipulate the different parts of latent code $w$ respectively. Multiple loss functions are used to constrain the synthetic image to fulfil the requirements.

## 3.2 PROBLEM FORMULATION

Our method leverages two latent space: $\mathcal{W}+$ space and CLIP-space, to transfer semantic-level facial attributes. For better expression, let $\mathcal{X}$ and $\mathcal{Y}$ denote the set of images and semantic features in CLIP-space, respectively. The image $x \in \mathcal{X}$ is inverted into corresponding latent code $w \in \mathcal{W}+$. $t$ denotes the input text prompt, with corresponding semantic feature $y_t$. Meanwhile, the parameters of pre-trained StyleGANv2 generator $G$ are frozen during training.

Given the text prompt $t$, the goal of facial attribute transfer model is to train a latent mapping network, which translates $w \backslash y$ to $\hat{w} \backslash \hat{y}$, and synthesizes the edited image $\hat{x}$ that meets the specific requirements. Our basic editing model can be formally defined as $\hat{x} = G(M(x, t))$, where $M$ is the manipulation network. Hence given the reference image $x_{ref} \in \mathcal{X}$, our semantic-aware attribute transfer model can be formally defined as $\hat{x} = G(M(x, x_{ref}, t))$.

## 3.3 DIRECTIONAL LATENT MAPPING

Only focusing on matching $\hat{y}$ with $y_t$ may cause irrelevant attribute changed. Therefore, we enforce the mapping network to focus on the change direction of semantic features in CLIP-space, and illustrate this idea in Figure 1. The details of the proposed directional latent mapping network is described as follows:

**Architecture.** It has been shown that the different layers of synthesis network, which corresponds to different parts of the latent code, control different granularity of semantic feature. In order to better exploit this property, we design the directional latent mapping network, and depict it in Figure 2. To a great extent, the degree of attributes disentanglement depends on the disentanglement of latent features. Therefore, we split the layers into $g$ groups, instead three (coarse, medium, and fine), with $g$ fully connected mapping networks, one for each group. The divided part of the latent code can be denoted as $w = [w_1, w_2, ..., w_g]$, so the mapping network is defined by:

$$M(w) = [M_1(w_1), M_2(w_2), ..., M_g(w_g)], \tag{2}$$

where $M_i$ is the $i$-th mapping network, and $[\cdot, \cdot]$ means concat operation. Then we use skip-connection operation to obtain the final manipulated latent code $\hat{w} = w + \alpha M(w)$, and feed it to the pre-trained StyleGANv2 generator $G$ to get the final manipulated facial image $\hat{x} = G(\hat{w})$.

**Training Objective.** The edited desired attribute is determined by the textual prompt $t$. Without paired training data, the mapping network could not correctly manipulate the latent code. In order to obtain extra powerful supervision, we use CLIP model to effectively extract semantic features (attributes) $y$ of the corresponding images and text:

$$\begin{aligned} y_i; \hat{y}_i &= E_I(x; \hat{x}), \\ y_t &= E_T(t), \end{aligned} \tag{3}$$

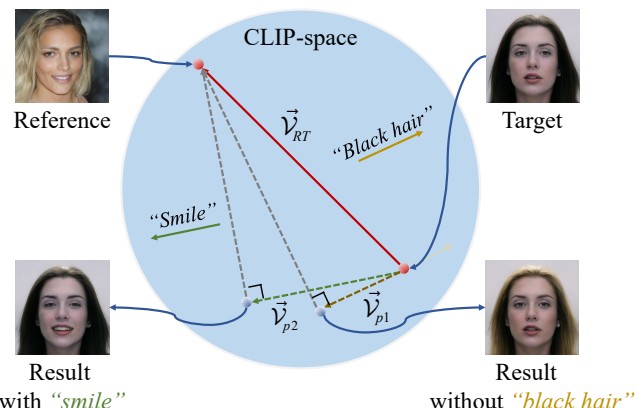

Figure 3: The illustration of attribute transfer loss $\mathcal{L}_{\text{AT}}$ in our Semantic Directional Decomposition Network (SDD-Net). We embed both the text prompt and reference\target face into CLIP-space. When the projected vector $\vec{\mathcal{V}}_p$ has the same direction with desired direction, the same attribute (e.g., smile) is transferred to the target face. Reversely, the opposite attribute (e.g., black hair) will be transferred when the direction is inverse.

where $E$ denotes the pre-trained multi-modal feature extractor integrated in CLIP, $y_i$ and $\hat{y}_i$ denotes the extracted semantic features of $x$ and $\hat{x}$, respectively. $y_t$ is the desired semantic feature extracted from textual prompt $t$.

In the latent CLIP-space, simply optimizing the matching degree between $\hat{y}_i$ and $y_t$ may cause irrelevant attribute to be changed. Therefore, rather than optimizing the matching degree to the utmost extent, we enforce the change direction between $y$ and $\hat{y}_i$ to align with the $y_t$, and propose the semantic direction consistency (SDC) loss. The SDC loss is given by:

$$
\begin{aligned}
\vec{I} &= \hat{y}_i - y_i, \\
\vec{T} &= y_t, \\
\mathcal{L}_{\text{SDC}} &= 1 - S(\vec{I}, \vec{T}),
\end{aligned}
\tag{4}
$$

where $S(\cdot, \cdot)$ is the similarity measurement. In this paper, we use the effective cosine similarity as the measurement.

The above objective can guide the mapping network to manipulate the latent code along the direction of textual prompt. In order to preserve the irrelevant parts, we use the following identity loss:

$$
\mathcal{L}_{\text{ID}} = 1 - \langle R(G(w)), R(G(\hat{w})) \rangle,
\tag{5}
$$

where $R$ is a pre-trained ArcFace (Deng et al., 2019) network for face recognition, and $\langle \cdot, \cdot \rangle$ computes the cosine similarity between its two arguments. Meanwhile, we use $L_2$ distance in $\mathcal{W}+$ space to control the degree of manipulation. Then, the whole training objective for directional latent mapping network is denoted as:

$$
\arg \min_{w \in \mathcal{W}+} \lambda_{\text{SDC}} \mathcal{L}_{\text{SDC}}(w, t) + \lambda_{L2} \|M(w)\|_2 + \lambda_{\text{ID}} \mathcal{L}_{\text{ID}}(w).
\tag{6}
$$

### 3.4 Semantic Directional Decomposition

Inspired by the above method, we found that a semantic attribute translation can be represented as a directional vector in CLIP-space. Based on this, we leverage the same architecture of directional mapping network, and propose a facial attribute transfer method via semantic-aware knowledge transfer, named as semantic directional decomposition network (SDD-Net).

Given the textual prompt $t$, we want to transfer the attribute, extracted from the reference image $x_{ref}$, to the target image $x_{tar}$. The $y_{ref}$ and $y_{tar}$ are their corresponding semantic features respectively. And the semantic feature of textual prompt $t$ is $y_t$.

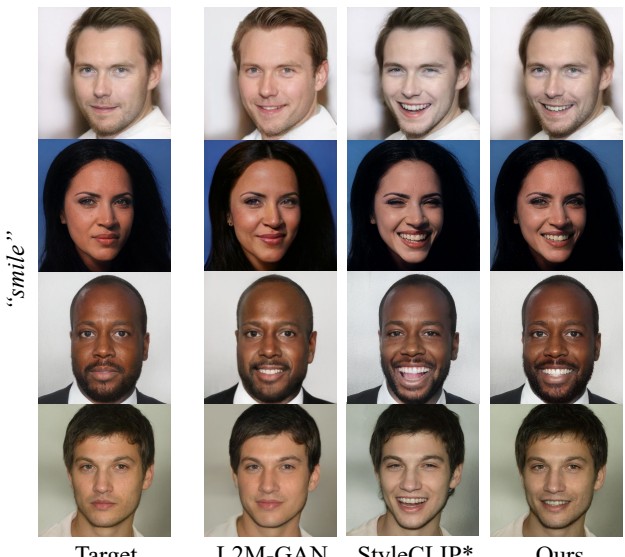

Figure 4: Qualitative results for facial attribute editing on attribute "smile". The left column are the target images sampled from CelebA-HQ dataset. The other column from left to right are the editing results of L2M-GAN (Yang et al., 2021a), StyleCLIP* (Patashnik et al., 2021), and our directional latent mapping network.

Due to the above direction-based manipulation, we can treat the facial attribute transfer problem as the attribute semantic feature projection problem. In the CLIP-space, a facial image can be treated as the composition of different semantic features. Therefore, we assume that different facial images can be converted to each other in semantic-level. In other words, one semantic feature can be translated to another semantic feature.

Under such an assumption, we first define the RT (reference-target) vector as: $\vec{\mathcal{V}_{RT}} = y_{ref} - y_{tar}$, which is the variance of two semantic features, and also is the direction of translation. We set the $y_t$ as the projection direction. In order to extract knowledge from the reference, we project RT vector $\vec{\mathcal{V}_{RT}}$ onto desired direction $\vec{\mathcal{V}_t} = y_t$:

$$\vec{\mathcal{V}_p} = \vec{\mathcal{V}_{RT}} \cdot \left( \frac{\vec{\mathcal{V}_t}}{|\mathcal{V}_t|} \right)^2, \tag{7}$$

where $\vec{\mathcal{V}_p}$ is the projected vector. Then we add it to the target semantic feature $y_{tar}$ to get the final goal:

$$y_{goal} = y_{tar} + \beta \vec{\mathcal{V}_p}, \tag{8}$$

where $\beta$ is the hyperparameter.

**Training Objective.** We use same architecture of directional latent mapping network to obtain the manipulated image $\hat{x} = G(M(\hat{w}))$, then match the semantic feature $\hat{y} = E_I(\hat{x})$ with $y_{goal}$ in CLIP-space. As illustrated in Figure 3, the attribute transfer loss $\mathcal{L}_{AT}$ is given by:

$$\mathcal{L}_{AT} = \mathrm{MSE}(\hat{y}, y_{goal}), \tag{9}$$

where $\mathrm{MSE}(\cdot, \cdot)$ is the mean squared error.

Meanwhile, we use the same $L2$ loss and identity loss to preserve the irrelevant parts. The whole training objective of SDD-Net is:

$$\underset{w \in \mathcal{W}+}{\arg\min} \ \lambda_{AT}\mathcal{L}_{AT} + \lambda_{L2}\mathcal{L}_2 + \lambda_{ID}\mathcal{L}_{ID}. \tag{10}$$

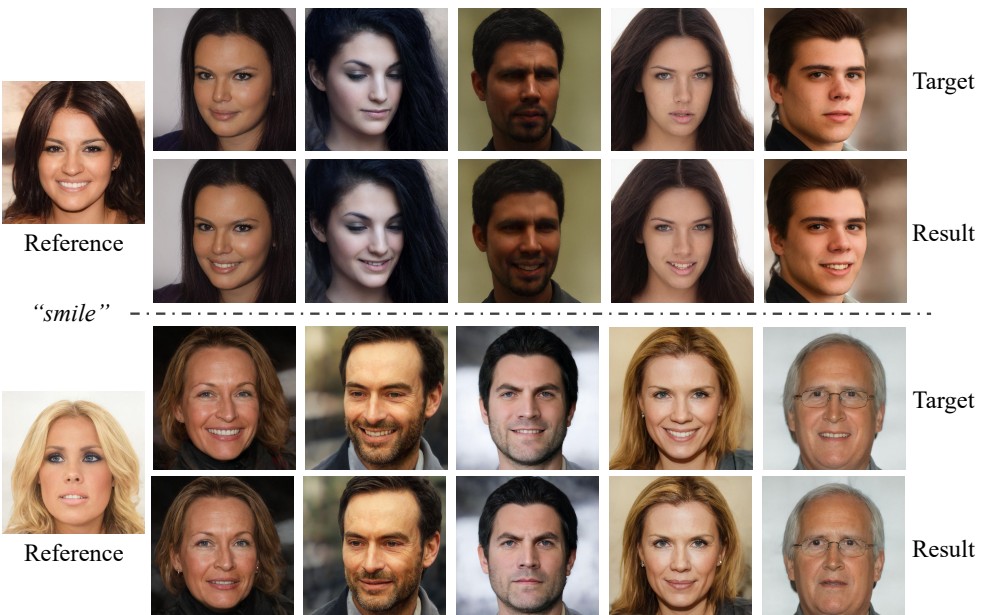

Figure 5: The qualitative results of our SDD-Net on single attribute transfer. Reference images in the left column. Target images in the first row of each parts, the rest is our manipulated images. Given the text prompt, our SDD-Net transfer the specific attribute ("smile") to the target images (top part) when the reference have the specific attribute. By contrast, the reverse attribute ("unsmiling") will be transferred (bottom part) when the specific attribute is not contained in the reference. Notice that, we only input one text prompt. So the SDD-Net could determine the forward or reverse transferring direction, according to the reference.

## 4 EXPERIMENTS

In this section, we first introduce the involved dataset and the implementation details. Then we will present the comparison results with several state-of-the-art facial attribute editing and facial attribute transfer methods to prove the effectiveness of our proposed method. Finally, the ablation studies will be presented to prove the effectiveness of our method.

### 4.1 DATASET

In order to achieve text-driven facial attribute editing and transfer, we choose the widely-used CelebA-HQ (Karras et al., 2017) dataset, which consists of 30,000 high quality facial images picked from the original CelebA (Liu et al., 2015) dataset. The size of each high quality image is 1024×1024. In the original dataset, each image has 40 attributes annotations inherited from the original CelebA. However in this work, we remove these annotations, and leverage CLIP model as powerful supervision. We also use the standard training, validation and test splits inherited from CelebA dataset.

### 4.2 IMPLEMENTATION DETAILS

In this subsection we provide the implementation details of our proposed networks. All images taken from the CelebA-HQ are inverted by e4e (Tov et al., 2021). We set the batch size and the number of total iterations to 5 and 50k respectively, during training. Our image editing is performed on StyleGANv2 pre-trained on FFHQ (Karras et al., 2019) dataset. We keep the StyleGANv2 generator fixed during training. The CLIP model is pre-trained on 400 million image-text pairs. We use the Vision Transformer (ViT) (Dosovitskiy et al., 2020) and a normal Transformer (Vaswani et al., 2017), which are integrated in CLIP model, as our image encoder and text encoder, respectively. Our directional latent mapping module is initialized using the pre-trained StyleGANv2, and trained

Reference    Target

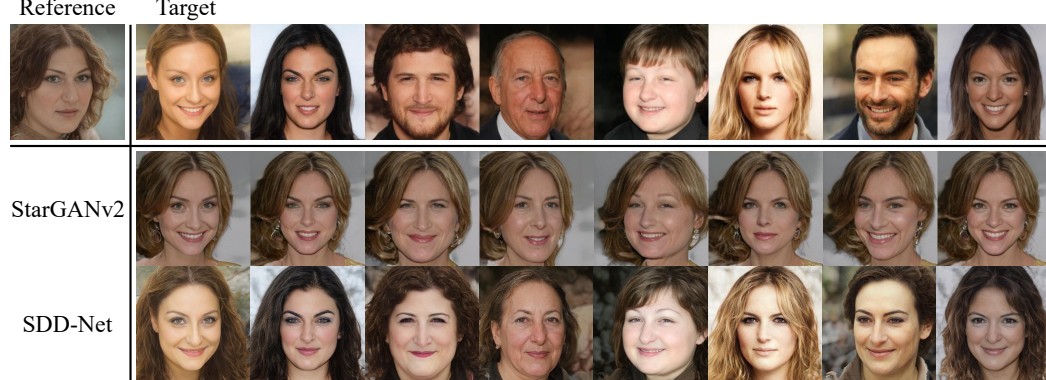

StarGANv2

SDD-Net

Figure 6: Experiments on reference-guided image synthesis on CelebA-HQ. Reference and target images both in the first row. The second and third raw are the synthetic results of StarGANv2 (Choi et al., 2020) and our SDD-Net respectively.

using Adam with the learning rate 5e-3. We set $g = 9$, $\alpha = 0.1$, and $\beta = 130$. For facial attribute editing, the hyperparameters are empirically set as $\lambda_{SDC} = 1$, $\lambda_{L2} = 0.4$, and $\lambda_{ID} = 0.02$. For facial attribute transfer, the hyperparameters are empirically set as $\lambda_{AT} = 1$, $\lambda_{L2} = 0.2$, and $\lambda_{ID} = 0.02$. Our methods are trained on PyTorch with a single TITAN RTX GPU. We optimize training objective through gradient descent, by back-propagating the gradient through the pre-trained and fixed StyleGAN generator $G$ and CLIP multi-modal encoder $E$.

### 4.3 BASELINE METHODS

**Facial Attribute Editing Methods.** We first compare our directional latent mapping network with the state-of-the-art facial attribute editing methods (i.e., L2M-GAN (Yang et al., 2021a), and Style-CLIP (Patashnik et al., 2021)) for the specific attribute: "smile". Due to the requirements of high-level semantic-aware knowledge, the smile attribute has become one of the most challenging global attributes. For L2M-GAN, we set the attribute domains as two ("smile" and "sad"), leveraging the domain label to guide the manipulation. For StyleCLIP, we use the latent mapper network as the baseline model, which is marked as StyleCLIP*.

**Facial Attribute Transfer Methods.** The StarGANv2 (Choi et al., 2020) learns to transform a source image reflecting the style of a given reference image. We compare our SDD-Net with Star-GANv2 method on reference-guided image synthesis, which is a challenging facial attribute transfer task. Reference-guided image synthesis requires that various high-level attributes in the reference images, such as hairstyle, makeup, beard and expression, should be transferrd to the target images, while the irrelevant information such as pose and identity should be preserved. For fair comparison, we resize the images to 256×256 in reference-guided image synthesis.

### 4.4 RESULTS AND ANALYSIS

**Facial Attribute Editing.** The qualitative results are shown in Figure 4. After careful comparison, we have following observations: (1) The L2M-GAN performs well at disentangling attributes during editing. However, due to the strong constrain of orthogonal loss, the manipulated attribute is not obvious. (2) Although the StyleCLIP* method can make the best use of semantic knowledge of CLIP-space, the irrelevant attributes are changed in facial attribute editing without correct guidance direction. (3) Our directional latent mapping network manipulates the attribute correctly and naturally, which demonstrates that the proposed semantic directional consistency (SDC) loss could enforce the editing model to change specific attributes while preserving irrelevant parts. It also proves that there exists latent directions corresponding to different semantic properties in latent space. By manipulating latent code along such direction or its opposite direction, we can add or remove attributes.

**Facial Attribute Transfer.** Inspired by the above direction-based latent space manipulation for facial attribute transfer, we leverage this peculiarity for facial attribute transfer task.

We first execute our SDD-Net on the "smile" attribute for single attribute transfer, and input the corresponding text prompt to the model. The qualitative results are shown in Figure 5. We observe that when reference has the consistent attribute appointed by the prompt, the SDD-Net could correctly transfer the specific attribute to the target face with irrelevant information preserved, as shown in the top part. On the contrary, when the reference has the opposite attribute, our SDD-Net also could transfer opposite attribute to the target without extra guidance. By fully leveraging the knowledge of CLIP-space, our SDD-Net could find the semantic-aware latent direction in the latent space. The experiments show the excellent performance of our SDD-Net.

For the reference-guide image synthesis, we set the projected vector $\vec{\mathcal{V}_p}$ is equal to the RT vector $\vec{\mathcal{V}_{RT}}$. Figure 6 provides qualitative comparison of the results. We observe that StarGANv2 mothed synthesizes images with a same style code. However, the results show that StarGANv2 model manipulates images in a limited space. As a result, the attributes of the synthetic faces tend to be exactly alike. In addition, the expression attribute such as "smile" is ignored during transferring. Compared to StarGANv2, our SDD-Net transfers the multiple meaningful attributes to each of the target faces in semantic-level. Meanwhile, our method manipulates the image along the latent direction leveraging the knowledge of CLIP in latent space, which allows our method to synthesize realistic facial images, rather than simple style transfer.

## 4.5 ABLATION STUDY

We conduct a qualitative ablation study, as shown in Figure 7, and show the significance of identity loss. We observe that, when hyperparameter $\lambda_{\text{ID}} = 0.1$, the ID loss hinders the attribute transfer. Then we experiment with $\lambda_{\text{ID}} = 0$, the attribute could be correctly transferred, but the identity information is changed. To trade-off, we set the $\lambda_{\text{ID}} = 0.02$.

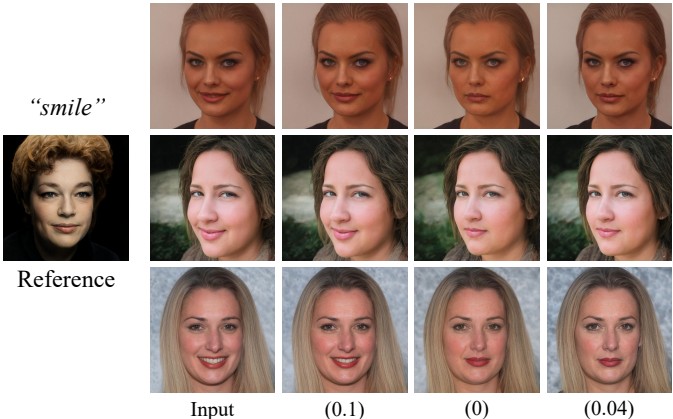

Figure 7: The ablation study of identity loss. Under each column we specify ($\lambda_{\text{ID}}$) identity loss. Obviously the ID loss is significant for facial attribute transfer.

## 5 CONCLUSIONS

In this paper, we first propose directional latent mapping network for text-driven facial attribute editing. By leveraging semantic direction consistency (SDC) loss, the directional latent mapping network could correctly edit relevant attribute while preserving irrelevant attributes. And on this basis, we propose semantic directional decomposition network (SDD-Net) for text-driven facial attribute transfer, which correctly transfers the semantic-aware attributes of reference image to the target image. Experiments show that our method achieves impressive performance on CelebA-HQ dataset.

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
