# OpenReview forum: "Text-Driven Image Manipulation via Semantic-Aware Knowledge Transfer"
_ICLR.cc/2022/Conference — ICLR 2022 Submitted_

### Official Review · Reviewer_V8mm · 2021-11-02

**Correctness:** 4
**Technical Novelty And Significance:** 2
**Empirical Novelty And Significance:** 2
**Recommendation:** 6
**Confidence:** 3

**Main Review:**

Strengths:
-- the proposed semantic directional decomposition strategy seems interesting and novel.

-- the experimental results reported in this submission seem better than prior works such as StyleCLIP.

Weakness:
-- more experiments with various attribute transfer are expected to further demonstrate the efficacy of this proposed method. In this submission, only a few attributes are considered ("black hair", "bang", "smile").

-- it seems that this proposed work can only handle one single attribute change/transfer? can it work when multiple attributes need to be transferred?

**Summary Of The Paper:**

This submission proposes a novel test-drivern directional latent mapping network with a new semantic direction consistency constraint for semantic-level facial attribute transfer. By treating the facial attribute transfer problem as an attribute semantic feature projection problem, a semantic directional decomposition strategy is proposed.

**Summary Of The Review:**

This submission proposes a novel test-drivern directional latent mapping network with a new semantic direction consistency constraint for semantic-level facial attribute transfer. By treating the facial attribute transfer problem as an attribute semantic feature projection problem, a semantic directional decomposition strategy is proposed. The reported experimental results seem good, especially comparing to the SOTA like StyleCLIP.

I have some concerns about the experiment settings and I feel some more discussions are needed. On one hand, only a few attributes are considered in the current submission, more experiments with various attribute transfer are expected; on the other hand, it is expected to know whether this method can still work when more than one attributes need to be transferred.

---

> ### Author Response · Authors · 2021-11-22
> **Thanks for your review!**
>
> The “smile” attribute is one of the challenging attributes. In this paper, we focus on the method that transfer the attributes in semantic-aware, by simply leveraging “smile” as a typical example. So the editing method of the other attributes is the same as that of “smile”. We also show some qualitative results in the supplementary material.

---

### Official Review · Reviewer_ksFh · 2021-11-02

**Correctness:** 3
**Technical Novelty And Significance:** 2
**Empirical Novelty And Significance:** Not applicable
**Recommendation:** 5
**Confidence:** 4

**Main Review:**

Strengths:
The idea and method are intuitive and easy to follow. Figures are pretty easy to understand.
The selected results are impressive.

Weaknesses:
Little novelty and confusing statements. The main contribution is essentially a loss term that measures latent similarity/distance in CLIP latent space. The other parts are from existing works, including StyleGANv2, e4e, and CLIP. Moreover, it seems unreasonable that text feature y_t can be treated as a semantic direction. Besides, the role of the reference image in attribute transfer is confusing since the text prompt plays a significant role in the presented results.
Insufficient evalutions. The authors have not justified whether their method works on a wide range of attributes with quantitative evaluations.
Hard to use in practice. Based on my understanding, the model needs re-training for different attributes and different controlling degrees, making it hard to use in practice.

**Summary Of The Paper:**

This paper proposes a new loss function for unsupervised facial attribute editing and transfer. Specifically, a latent mapping network is trained by optimizing the similarity/distance between the generated and desired image in the CLIP latent space. Experiments show some comparisons and ablations for the "smile" attribute.

**Summary Of The Review:**

I think this paper has limited novelty and insufficient evaluations. The detailed comments are listed above.

---

### Official Review · Reviewer_nZQ1 · 2021-11-03

**Correctness:** 3
**Technical Novelty And Significance:** 2
**Empirical Novelty And Significance:** 2
**Recommendation:** 3
**Confidence:** 4

**Main Review:**


strengths:
1. The incorporation of attribute-aware mechanism based on StyleCLIP is technically sound.
2. Qualitative results are provided, demonstrating the proposed method performs fine on CelebA-HQ dataset.

weaknesses:
1. The comparison to StyleCLIP is not sufficient, especially considering this paper is a follow-up work of StyleCLIP and there are no quantitative experiments to support the authors' claims. For instance, only 4 examples in Figure 4 demonstrate the compairson to StyleCLIP. From my view, the outputs of StyleCLIP and the proposed method don't have a big difference. If the authors would like to claim that their method can edit relevance and keep irrelevance while StyleCLIP cannot, it would be better to point out in the Figure which irrelevant parts of faces are changed by StyleCLIP. I would recommend the authors include at least 20 images in the supplementary to demonstrate the effectiveness of their approach. (I didn't find the supplementary materials in the current version)
2. The paper only conducts experiments on CelebA-HQ dataset and the title is "text-driven image manipulation" rather than "text-driven celebrity face manipulation. I would recommend the authors perform experiments on more datasets such as cars dataset [1] or dogs dataset [2] following StyleCLIP.
3. Typos like "definded" in the paper should be corrected. Besides, math symbols should be explained in advance. For example, the mapping network "M" is first mentioned in Section 3.2 and the authors didn't point out the definition of "M" until Section 3.3.


[1]Fisher Yu, Ari Seff, Yinda Zhang, Shuran Song, Thomas Funkhouser, and Jianxiong Xiao. Lsun: Construction of a large-scale image dataset using deep learning with humans in the loop. arXiv preprint arXiv:1506.03365, 2015
[2] Yunjey Choi, Youngjung Uh, Jaejun Yoo, and Jung-Woo Ha. StarGAN v2: Diverse image synthesis for multiple domains. In Proc. CVPR, pages 8188–8197, 2020.



**Summary Of The Paper:**

This paper proposes a latent mapping mechanism based on StyleCLIP to disentangle the semantic attributes of human face. The authors provide some qualitative results and ablation study on CelebA-HQ dataset.

**Summary Of The Review:**

Overall I tend to reject the paper. Please refer to the Main Review.

---

### Official Review · Reviewer_wMtJ · 2021-11-03

**Correctness:** 2
**Technical Novelty And Significance:** 3
**Empirical Novelty And Significance:** Not applicable
**Recommendation:** 5
**Confidence:** 3

**Main Review:**

Personally, the paper was enjoyable. It was intuitive and a clever idea, end-to-end. The figures depicting the methods and theory are informative; the experiments seem convincing, and the overall framework seems reproducible.


**Summary Of The Paper:**

The authors proposed a directional latent mapping network for facial attribute editing via text inputs. The directional latent mapping network could correctly edit relevant attributes while preserving irrelevant attributes via training with the semantic direction consistency (SDC) loss. This paved the way to a novel semantic directional decomposition network (SDD-Net) for text-driven facial attribute transfer: SDD-Net transfers semantic-aware attributes from reference images to a target, with the multi-modal approach guiding the process via text input descriptions. CelebA-HQ dataset was used to compare results with recent SOA methods.

**Summary Of The Review:**


After reading the paper for the first time, I figured a sure "accept." However, I then noticed they always used it for "smile," besides when "not smiles" (i.e., any attribute but "smiles" were transferred. Not that this should take everything away from the work. Nonetheless, I believe the authors are not transparent enough in this aspect of the paper-- do we need K models for K attributes? The authors attempted to show "black hair" and "bangs" but it was unclear what was important in the results.

They showed many image montages throughout: it is good to highlight the 'take-home message.' Too many times was I zooming in trying to understand the differences between samples. A better explanation of the variations is needed.

I cannot help but see this work as extremely high in potential: a potential that far surpasses the quality of the current version. More insight, edge cases, and failure modes would be a plus. More attribute types would be best. Which attributes work, which doesn't, and how does this relate to the motivation (i.e., global vs. local).

If this was FG I would suggest "accept," but as the paper shows plenty, but does not explain enough. It's only being used on smiles for most of the time, with no emphasis on this, just seemed suspect (not that it was intentional, but just was a letdown when I finished the first pass, was excited about the paper, and then realized this which was a total let down)..

---

> ### Author Response · Authors · 2021-11-22
> **Thanks for your review!**
>
> For the attributes in facial image, “smile” is one of the most representative attributes. In this paper, we focus on the method that transfer the attributes in semantic-aware, by simply leveraging “smile” as a typical example. Therefore, the editing method of the other attributes is the same as that of “smile”. You’re right, our method needs K models for K different attributes. We also show some qualitative results in the supplementary material.
>
> Our key hypothesis is that two different facial images can be semantically interchangeable. By leveraging a simple vector projection, we can isolate the desired attributes and transfer them to the target face. Thanks to the directivity of vectors, our method can automatically determine the relationship between the desired attributes and a given reference face, and transfer the corresponding attributes. Obviously, it does have a lot of limitations. It is surprising that the operation of vector projection can be efficient in CLIP space, a complex manifold space. The method requires further analysis. Besides, our method sometimes dose not disentangle local attributes well such as “color” attribute. We need stronger local constraints.
>
> We hope we have addressed most of your comments satisfactorily. Thank you.

---

> > ### Comment · Reviewer_wMtJ · 2021-11-23
> > **Rebuttal Response**
> >
> > It does so so. It would have been nice for this to be emphasized and explored more in the paper, but, considering the extensive experiments done for smile/not smile, I went up a tick on the decision scale. Best of luck :)

---

### Decision · Program_Chairs · 2022-01-20

**Decision:**

Reject

**Comment:**

This submission proposes a new loss function for facial attribute GAN editing and transfer via text inputs.
A latent mapping mechanism based on StyleCLIP is used to disentangle the semantic attributes of human face.
The resulting semantic directional decomposition network (SDD-Net) transfers attributes from reference image to a target guided with text descriptions. Experiments show on CelebA-HQ dataset some qualitative results and ablations  for the « smile » attribute.

The main contribution is essentially a loss term that measures latent similarity in CLIP latent space.
Most of the reviewers are not convinced by the approach and have raised several issues. One can question the relevancy of the way that text features are used (as a semantic direction). The role of the reference image in attribute transfer is is also questionable in the proposed framework.
Additionally, evaluation is not sufficient, in particular to investigate whether the proposed method works on a wide range of attributes.
The paper only conducts experiments on CelebA-HQ dataset. It would be interesting to have experiments on other  datasets.
The comparison to StyleCLIP is also insufficient, and there are no quantitative experiments to support the authors' claims.
We encourage the authors to take into account all these remarks and Rs' comments in order to get an improved proposition for a future conference.